# AutoBSS: An Efficient Algorithm for Block Stacking Style Search

**Yikang Zhang**
Huawei
zhangyikang7@huawei.com

**Jian Zhang**
Huawei
zhangjian157@huawei.com

**Zhao Zhong**
Huawei
zorro.zhongzhao@huawei.com

## Abstract

Neural network architecture design mostly focuses on the new convolutional operator or special topological structure of network block, little attention is drawn to the configuration of stacking each block, called Block Stacking Style (BSS). Recent studies show that BSS may also have an unneglectable impact on networks, thus we design an efficient algorithm to search it automatically. The proposed method, AutoBSS, is a novel AutoML algorithm based on Bayesian optimization by iteratively refining and clustering Block Stacking Style Coding (BSSC), which can find optimal BSS in a few trials without biased evaluation. On ImageNet classification task, ResNet50/MobileNetV2/EfficientNet-B0 with our searched BSS achieve 79.29%/74.5%/77.79%, which outperform the original baselines by a large margin. More importantly, experimental results on model compression, object detection and instance segmentation show the strong generalizability of the proposed AutoBSS, and further verify the unneglectable impact of BSS on neural networks.

## 1 Introduction

Recent progress in computer vision is mostly driven by the advance of Convolutional Neural Networks (CNNs). With the evolution of network architectures from AlexNet [1], VGG [2], Inception [3] to ResNet [4], the performance has been steadily improved. Early works [1, 2, 5] designed layer-based architectures, while most of the modern architectures [3, 4, 6, 7, 8, 9] are block-based. For those block-based networks, the design procedure consists of two steps: (1) designing the block structure. (2) stacking the blocks to construct a complete network architecture. The manner for stacking blocks is named as Block Stacking Style (BSS) inspired by BCS from [10]. Compared with the block structure, BSS draws little attention from the community.

The modern block-based networks are commonly constructed by stacking blocks sequentially. The backbone can be divided into several stages, thus BSS can be simply described by the number of blocks in each stage and the number of channels for each block. The general rule to set channels for each block is to double the channels when downsampling the feature maps. This rule is adopted by a lot of famous networks, such as VGG [2], ResNet [4] and ShuffleNet [11, 8]. As for the number of blocks in each stage, there is merely a rough rule that more blocks should be allocated in the middle stages [4, 7, 8]. Such human design paradigm arouses our questions: Is this the best BSS configuration for all networks? However, recent works show that BSS may have an unneglectable impact on the performance of a network [12, 10]. [12] find a kind of pyramidal BSS style which is better than the original ResNet. Even further, [10] tries to use reinforcement learning to find optimal

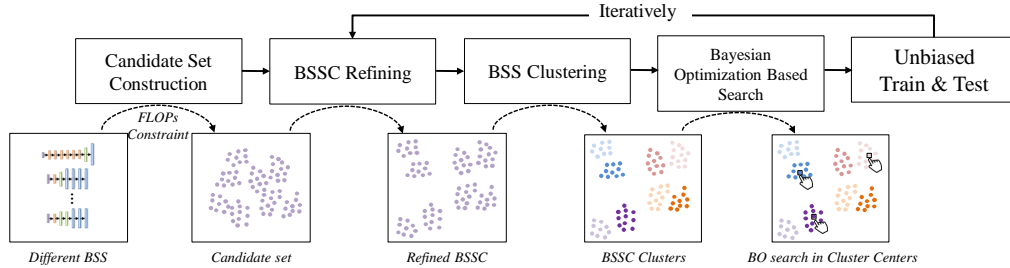

Figure 1: The overall framework of our proposed AutoBSS.

Block Connection Style (similar to BSS) for searched network block. These studies imply that the design of BSS has not been fully understood.

In this paper, we aim to break the BSS designing principles defined by human, and propose an efficient AutoML based method called **AutoBSS**. The overall framework is shown in Figure 1, where each BSS configuration is represented by Block Stacking Style Coding (BSSC). Our goal is to search an optimal BSSC with the best accuracy under some target constraints (e.g. FLOPs or latency). Current AutoML algorithms usually use a biased evaluation protocol to accelerate search [13, 14, 15, 16, 17], such as early stop or or parameter sharing. However, BSS search space has its unique benefits, where BSSC has a strong physical meaning. Each BSSC affects the computation allocation of a network, thus we have an intuition that similar BSSC may have similar accuracy. Based on this intuition, we propose a Bayesian Optimization (BO) based approach. However, BO based approach does not perform well in a large discrete search space. Benefit from the strong prior, we present several methods to improve the effectiveness and sample efficiency of BO on BSS search. *BSS Clustering* aggregates BSSC into clusters, each BSSC in the same cluster have similar accuracy, thus we only need to search over cluster centers. *BSSC refining* enhances the coding representation by increasing the correlation between BSSC and corresponding accuracy. To improve BSS Clustering, we propose a candidate set construction method to select a subset from search space efficiently. Based on these improvements, AutoBSS is extremely sample efficient and only needs to train tens of BSSC, thus we use an *unbiased evaluation scheme* and avoid the strong influence caused by widely used tricks in neural architecture search (NAS) methods, such as early stopping or parameter sharing.

Experiment results on various tasks demonstrate the superiority of our proposed method. The BSS searched within tens of samplings can largely boost the performance of well-known models. On ImageNet classification task, ResNet50/MobileNetV2/EfficientNet-B0 with searched BSS achieve 79.29%/74.5%/77.79%, which outperform the original baselines by a large margin. Perhaps more surprisingly, results on model compression(+1.6%), object detection(+0.91%) and instance segmentation(+0.63%) show the strong generalizability of the proposed AutoBSS, and further verify the unneglectable impact of BSS on neural networks.

The contributions of this paper can be summarized as follows:

- We demonstrate that BSS has a strong impact on the performance of neural networks, and the BSS of current state-of-the-art networks is not the optimal solution.

- We propose a novel algorithm called AutoBSS that can find a better BSS configuration for a given network within only tens of trials. Due to the sample efficiency, AutoBSS can search with unbiased evaluation under limited computing cost, which overcomes the errors caused by the biased search scheme of current AutoML methods.

- The proposed AutoBSS improves the performance of widely used networks on classification, model compression, object detection and instance segmentation tasks, which demonstrate the strong generalizability of the proposed method.

## 2 Related Work

### 2.1 Convolutional Neural Network Design

The convolutional neural networks (CNNs) have been applied in many computer vision tasks [18, 1]. Most of modern network architectures [3, 4, 6, 7, 8] are block-based, where the design process is usually two phases: (1) designing a block structure, (2) stacking blocks to form the complete structure, in this paper we call the second phase BSS design. Many works have been devoted to effective and efficient block structure design, such as bottleneck [4], inverted bottleneck [7] and shufflenet block [8]. However, little effort has been made to BSS design, which has an unneglectable impact on network performance based on recent studies [12, 10]. There are two commonly-used rules for designing BSS: (1) doubling the channels when downsampling the feature maps, (2) allocating more blocks in the middle stages. These rough rules may not make the most potential of a carefully designed block structure. In this paper, we propose an automatic BSS search method named AutoBSS, which aims to break the human-designed BSS paradigm and find the optimal BSS configuration for a given block structure within a few trials.

### 2.2 Neural Architecture Search

Neural Architecture Search has drawn much attention in recent years, various algorithms have been proposed to search network architectures with reinforcement learning [19, 13, 20, 21, 22], evolutionary algorithm [23, 24], gradient-based method [25, 26] or Bayesian optimization-based method [27]. Most of these works [13, 20, 25, 24] focus on the micro block structure search, while our work focuses on the macro BSS search when the block structure is given. There are a few works related to BSS search [28, 9]. Partial Order Pruning (POP) [28] samples new BSS randomly while utilizing the evaluated BSS to prune the search space based on Partial Order Assumption. However, the search space after pruning is still too large, which makes it difficult to search BSS by random sampling. EfficientNet [9] simply grid searches three constants to scale up the width, depth, and resolution. BlockQNN [10] uses reinforcement learning to search BSS, however, it needs to evaluate thousands of BSS and uses early stop training tricks to reduce time cost. OnceForAll [29] uses weight sharing technique to progressively search the width and the depth of a supernet. The aforementioned methods are either sample inefficient or introduce some biased tricks for evaluation, such as early stop or weight sharing. Note that these tricks affect the search performance strongly [30, 31], where the correlation between the final accuracy and searched accuracy is very low. Different from those methods, our proposed AutoBSS uses an unbiased evaluation scheme and utilizes an efficient Bayesian Optimization based search method with BSS refining and clustering to find an optimal BSS within tens of trials.

## 3 Method

Given a building block of a neural network, BSS defines the number of blocks in each stage and channels for each block, which can be represented by a fixed-length coding, namely Block Stacking Style Coding (BSSC). BSSC has a strong physical meaning that describes the computation allocation in each stage. Thus we have a prior that similar BSSC may have similar accuracy. To benefit from this hypothesis, we propose an efficient algorithm to search BSS by Bayesian Optimization. However, BO based method does not perform well in a large discrete search space. To address this problem, we propose BSS Clustering to aggregate BSSC into clusters, and we only need to search over cluster centers efficiently. To enhance the BSSC representation, we also propose BSSC Refining to increase the correlation between coding and corresponding accuracy. Moreover, as the search space is usually huge, to perform BSS clustering efficiently, we propose Candidate Set Construction method to select a subset effectively. We will introduce these methods in the following subsections in detail.

### 3.1 Candidate Set Construction

The goal of AutoBSS is to search an optimal BSSC under some target constraints (e.g. FLOPs or latency). We denote the search space under the constraint as $\Lambda$. Each element of $\Lambda$ is a BSSC, denoted as $x$, with the $i$-th element as $x_i$ and the first $i$ elements as $x_{[:i]}$, $i = 0, ..., m$. The set of possible values for $x_i$ is represented as $C^i = \{c_0^i, c_1^i, ...\}$, thus $x_{[:i+1]} = x_{[:i]} \cup c_j^i, c_j^i \in C^i$. In most cases, $\Lambda$

is too large to enumerate, and clustering using the full search space is infeasible. Thus we need to select a subset of $\Lambda$ as the candidate set $\Omega$.

To make the BSS search in this subset more effectively, $\Omega$ aims to satisfy two criterions: (1) the candidates in $\Omega$ and $\Lambda$ should have similar distributions. (2) the candidates in $\Omega$ should have better accuracy than the unpicked ones in the search space.

To make the distribution of candidates in $\Omega$ similar with $\Lambda$, an intuitive way is to construct $\Omega$ via random sampling in $\Lambda$. During each sampling, we can sequentially determine the value of $x_0, ..., x_m$. The value of $x_i$ is selected from $C^i$, where $c_j^i$ corresponds with the possibility $P_j^i$. It can be proved in random sampling that,

$$P_j^i = \frac{|S(x_{[:i]} \cup c_j^i)|}{\sum_{\hat{j}}(|S(x_{[:i]} \cup c_{\hat{j}}^i)|)}, \ where \ S(x_{[:r]}) = \{\hat{x}|\hat{x} \in \Lambda, \hat{x}_{[:r]} = x_{[:r]}\}. \tag{1}$$

However, we can not get the value of $|S(x_{[:r]})|$ because it needs to enumerate each element in $\Lambda$. Thus, we simply utilize the approximate value $|S^{\mathcal{D}=0}(x_{[:r]})|$ by the following equation 2 recursively, where $\mathcal{D}$ denotes the recursion depth, $c_{mid}^r$ denotes the median of $C^r$.

$$|S^{\mathcal{D}=d}(x_{[:r]})| = \begin{cases} \sum_{j}\{|S^{\mathcal{D}=d+1}(x_{[:r]} \cup c_j^r)|\}, & if \ d \leq 2 \\ |S^{\mathcal{D}=d+1}(x_{[:r]} \cup c_{mid}^r)| \times |C^r|, & if \ d > 2 \end{cases} \tag{2}$$

To make the candidates in $\Omega$ have better potential than the unselected ones in $\Lambda$, we post-process each candidate $x \in \Omega$ in the following manner. We first randomly select one dimension $x_i$, then increase it by a predefined step size if this doesn't result in larger FLOPs or latency than the threshold. This process is repeated until no dimension can be increased. Increasing any dimension $x_i$ means increasing the number of channels or blocks, as shown in works like [7, 8, 12], it necessarily makes the resulting network perform better.

## 3.2  BSSC Refining

To demonstrate the correlation between BSSC and accuracy, we randomly sample 220 BSS for ResNet18 and evaluate them on ImageNet classification task. To make the distance between BSSC reasonable, we firstly standardize each dimension individually, i.e. replacing each element with Z-score. This procedure can be regarded as the first refining. Then, we show the relationship between Euclidean distance and accuracy discrepancy (the absolute difference of two accuracies) in Figure 2.

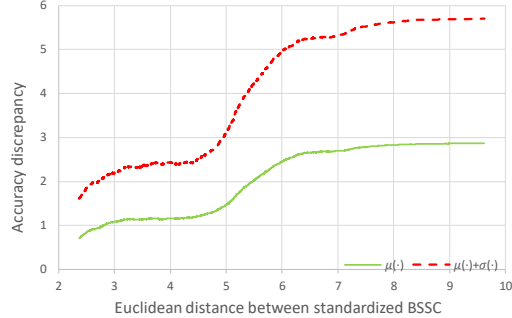

It can be observed that accuracy discrepancy tends to increase when distance gets larger, this phenomenon verifies our intuition that similar BSSC has similar accuracy. However the correlation is nonlinear. Because the Bayesian Optimization approach is based on Lipschitz-continuous assumption [32], we should refine it to a linear correlation so that the assumption

Figure 2: The relationship between BSSC distance and accuracy discrepancy. $\mu(\cdot)$ and $\sigma(\cdot)$ represent mean value and standard deviation respectively.

can be satisfied better. Therefore, we utilize evaluated BSS to refine BSSC from the second search iteration. We simply use a linear layer to transform it in this work. We set the initial weight of this linear layer as an identity matrix, and train the model with the following loss,

$$Loss_{dy} = \left(\frac{|y^{(0)} - y^{(1)}|}{|y^{(2)} - y^{(3)}|} - \frac{\|\hat{x}^{(0)} - \hat{x}^{(1)}\|_{L_2}}{\|\hat{x}^{(2)} - \hat{x}^{(3)}\|_{L_2}}\right)^2, \tag{3}$$

where $\hat{x}^{(0)}, ..., \hat{x}^{(3)}$ are transformed from four randomly selected evaluated BSSC and $y^{(0)}, ..., y^{(3)}$ are the corresponding accuracies.

### 3.3 BSS Clustering

After refining, neighboring BSSC naturally corresponds with similar accuracies. Thus searching from the whole candidate set is not necessary. We aggregate BSSC into clusters and search from the cluster centers efficiently. Besides, it brings two extra benefits. Firstly, it helps to avoid the case that all BSSC are sampled from a local minimum. Secondly, it makes the sampled BSSC dispersed enough, thus the GP model built on which can better handle the whole candidate set.

We adopt the k-means algorithm [33] with Euclidean distance to aggregate the refined BSSC. The number of clusters will increase during each iteration, while we only select the same number of BSSC from cluster centers in each iteration. It is because the refined BSSC can measure the accuracy discrepancy more precisely and the GP model becomes more reliable with the increase of evaluated BSSC.

### 3.4 Bayesian Optimization Based Search

We build a model for the accuracy as $f(x)$ based on GP. Because the refined BSSC has a relatively strong correlation with accuracy, we simply use a simple kernel $\kappa(x, x') = exp(-\frac{1}{2\sigma^2}\|x - x'\|_{L_2}^2)$. We adopt expected improvement (EI) as the acquisitions. Given $\mathcal{O} = \{(x^{(0)}, y^{(0)}), (x^{(1)}, y^{(1)}), ...(x^{(n)}, y^{(n)})\}$, where $x^{(i)}$ is a refined BSSC for an evaluated BSS, $y^{(i)}$ is the corresponding accuracy. Then,

$$\varphi_{EI}(x) = \mathbb{E}(max\{0, f(x) - \tau | \mathcal{O}\}), \tau = \max_{i \leq n} y^{(i)}. \tag{4}$$

To reduce the time consumption and take advantage of parallelization, we train several different networks at a time. Therefore, we use the expected value of EI function (EEI, [34]) to select a batch of unevaluated BSSC from cluster centers. Supposing $x^{(n+1)}, x^{(n+2)}, ...$ are BSSC for selected BSS with unknown accuracies $\hat{y}^{(n+1)}, \hat{y}^{(n+2)}, ...,$ thus

$$\varphi_{EEI}(x) = \mathbb{E}(\mathbb{E}(max\{0, f(x) - \tau | \mathcal{O}, (x^{(n+1)}, \hat{y}^{(n+1)}), (x^{(n+2)}, \hat{y}^{(n+2)}), ...\})), \tag{5}$$

here $\hat{y}^{(n+j)}, j = 1, 2, ...,$ is a variable of Gaussian distribution with mean and variance depend on $\{\hat{y}^{(n+k)} | 1 \leq k < j\}$. The value of equation 5 is calculated by Monte Carlo simulations [34] at each cluster center, the one with the largest value will be selected. More details are illustrated in Appendix A.1.

## 4 Experiments

In this section, we conduct the main experiments of BSS search on ImageNet classification task [35]. Then we conduct experiments to analyze the effectiveness of BSSC Refining and BSS Clustering. Finally, we extend the experiments to model compression, detection and instance segmentation to verify the generalization of AutoBSS. The detailed settings of experiments are demonstrated in the Appendix B.

### 4.1 Implementation Details

**Target networks and Training hyperparameters** We use ResNet18/50 [4], MobileNetV2 [7] and EfficientNet-B0/B1 [9] as target networks, and utilize AutoBSS to search a better BSS configuration for corresponding networks under the constraint of FLOPs. The detailed training settings are shown in Appendix B.1.

**Definition of BSSC** We introduce the definition of BSSC for EfficientNet-B0/B1 as an example, others are introduced in Appendix B.2. The main building block of EfficientNet-B0 is MBConv [7], but swish [36] and squeeze-and-excitation mechanism [37] are added. Then EfficientNet-B1 is constructed by grid searching three constants to scale up EfficientNet-B0. EfficientNet-B0/B1 consists of 9 stages, the BSSC is defined as the tuple $\{C_3, ..., C_8, L_3, ..., L_8, T_3, ..., T_8\}$, $C_i$, $L_i$ and $T_i$ denote the output channels, number of blocks and expansion factor[7] for stage $i$, respectively.

Table 1: Single crop Top-1 accuracy (%) of different BSS configurations on ImageNet dataset

| Method | FLOPs | Params | Impl. (120 ep.) | Ref. | Impl. (350 ep.) |
|---|---|---|---|---|---|
| ResNet18 | 1.81B | 11.69M | 71.21 | 69.00[28]) | 72.19 |
| ResNet18$^{Rand}$ | 1.74B | 24.87M | 72.34 | - | - |
| ResNet18$^{AutoBSS}$ | 1.81B | 16.15M | 73.22 | - | 73.91 |
| ResNet50 | 4.09B | 25.55M | 77.09 | 76.00[9] | 77.69 |
| ResNet50$^{Rand}$ | 3.69B | 23.00M | 77.48 | - | - |
| ResNet50$^{AutoBSS}$ | 4.03B | 23.73M | 78.17 | - | 79.29 |
| MobileNetV2 | 300M | 3.50M | 72.13 | 72.00[7] | 73.90 |
| MobileNetV2$^{Rand}$ | 298M | 4.00M | 72.13 | - | - |
| MobileNetV2$^{AutoBSS}$ | 296M | 3.92M | 72.96 | - | 74.50 |
| EfficientNet-B0 | 385M | 5.29M | - | 77.10[9] | 77.12 |
| EfficientNet-B0$^{Rand}$ | 356M | 6.67M | - | - | 76.73 |
| EfficientNet-B0$^{AutoBSS}$ | 381M | 6.39M | - | - | 77.79 |
| EfficientNet-B1 | 685M | 7.79M | - | 79.10[9] | 79.19 |
| EfficientNet-B1$^{Rand}$ | 673M | 10.19M | - | - | 78.56 |
| EfficientNet-B1$^{AutoBSS}$ | 684M | 10.17M | - | - | 79.48 |

**Detail Settings of AutoBSS Searching**  We use FLOPs of original BSS as the threshold for Candidate Set Construction. The candidate set $\Omega$ always has 10000 elements in all experiments. The number of iterations is set as 4, during each iteration 16 BSSC will be evaluated, so that in total *only 64 networks* will be trained in the searching process. Benefit from the sample efficiency, we use an *unbiased evaluation scheme* for searching, namely each candidate is trained fully without early stopping or parameter sharing. We train 120 epochs for ResNet18/50 and MobileNetV2, 350 epochs for EfficientNet-B0/B1. We set the number of clusters as 16, 160, $\frac{N}{10}$ and $N$ for each iteration, here $N$ denotes the size of candidate set $\Omega$. As indicated in [38], random search is a hard baseline to beat for NAS. Therefore we also randomly sample 64 BSSC from the same search space as a baseline for each network.

## 4.2  Results and Analysis

The results of ImageNet are shown in Table 1. More details are shown in Appendix B.3. Compared with the original ResNet18, ResNet50 and MobileNetV2, we improve the accuracy by **2.01**%, **1.08**% and **0.83**% respectively. It indicates BSS has an unneglectable impact on the performance, and there is a large improvement room for the manually designed BSS.

EfficientNet-B0 is developed by leveraging a reinforcement learning-based NAS approach [20, 9], BSS is involved in their search space as well. Our method achieves 0.69% improvement. The reinforcement learning-based approach needs tens of thousands of samplings while our method needs only 64 samplings, which is much more efficient. In addition, the 0.38% improvement on EfficientNet-B1 demonstrates the superiority of our method over grid search, which indicates that AutoBSS is a more elegant and efficient tool for scaling neural networks.

ResNet18/50$^{Rand}$, MobileNetV2$^{Rand}$ and EfficientNet-B0/B1$^{Rand}$ in Table 1 are networks with the randomly searched BSS, the accuracy for them is 0.88/0.69%, 0.83% and 1.06/0.92% lower compared with our proposed AutoBSS. It indicates that our method is superior to the hard baseline random search for NAS [38].

We also visualize the searching process of ResNet18 in Figure 3 (a), where each point represents a BSS. The searching process consists of 4 iterations, during each iteration, 16 evaluated BSS will be sorted based on the accuracy for better visualization. From the figure, we have two observations:

1) The searched BSS within the first iteration is already relatively good. It mainly comes from two points. Firstly, Candidate Set Construction excludes a large number of BSS which are expected to have a bad performance. Secondly, BSS Clustering helps to avoid the case that all BSS are sampled from a local minimum.

2) The best BSS is sampled during the last two iterations. It is because the growing number of evaluated BSS makes the refined BSSC and GP model more effective. As for why the best BSS is not always sampled during the last iteration, it is because we adopt EI acquisition function [32], which focuses on not only exploitation but also exploration.

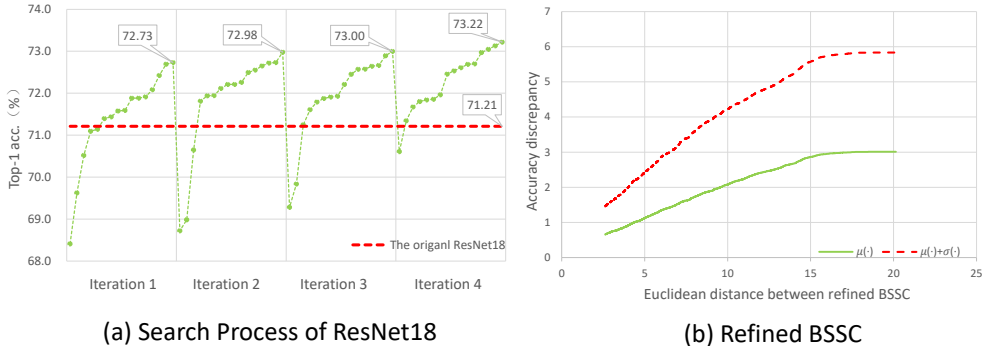

(a) Search Process of ResNet18          (b) Refined BSSC

Figure 3: (a) The searching process of ResNet18. (b) Correlation between refined BSSC and accuracy

## 4.3 Analysis for BSSC Refining and BSS Clustering

To demonstrate the effectiveness of BSSC Refining, we use the same BSS for ResNet18 as section 3.2 to plot relations between refined BSSC distance and network accuracy in Figure 3(b).The linear model for refining is trained with 16 randomly selected BSSC and makes the mapping from refined BSSC to accuracy satisfy the Lipschitz-continuous assumption of Bayesian Optimization approach [32]. That is, there exists constant $C$, such that for any two refined BSSC $x_1$, $x_2$: $|Acc(x_1) - Acc(x_2)| \leq C\|x_1 - x_2\|$. The red dashed line in Figure 2 and Figure 3(b) (mean value plus standard deviation) can be regarded as the upper bound of accuracy discrepancy $|Acc(x_1) - Acc(x_2)|$. After refined with the linear model, it becomes much more close to the form of $C\|x_1 - x_2\|$.

To prove the effectiveness of BSS Clustering, we simply carry out an experiment to search the BSS of ResNet18 without BSS Clustering. We compare the best 5 BSS in Table 2. It can be observed that the accuracy drops significantly without the BSS clustering.

Table 2: The best 5 BSS searched with/without BSS Clustering.

|  | Without BSS Clustering | With BSS Clustering |
|---|---|---|
| Top-1 acc. (%) | 72.56 (0.42 ↓) | 72.98 |
|  | 72.59 (0.41 ↓) | 73.00 |
|  | 72.60 (0.45 ↓) | 73.05 |
|  | 72.63 (0.50 ↓) | 73.13 |
|  | 72.70 (0.52 ↓) | 73.22 |
| Mean(%) | 72.62 (0.46 ↓) | 73.08 |

## 4.4 Generalization to Model Compression

Model compression aims to obtain a smaller network based on a given architecture. By adopting a smaller FLOPs threshold, model compression can be achieved with our proposed AutoBSS as well. We conduct an experiment on MobileNetV2, settings are identical with section 4.1 except for FLOPs threshold and training epochs. We compare our method with Meta Pruning [39], ThiNet [40] and Greedy Selection [41]. The results are shown in Table 3, AutoBSS improves the accuracy by a large margin. It indicates that pruning based methods is less effective than scaling a network using AutoBSS.

Table 3: Compared with other methods on MobileNetV2.

| Method | FLOPs | Params | Top-1 acc. (%) |
|---|---|---|---|
| Uniformly Rescale | 130M | 2.2M | 68.05 |
| Meta Pruning [39] | 140M | - | 68.20 [39] |
| ThiNet [40] | 175M | - | 68.60 [41] |
| Greedy Selection [41] | 137M | 2.0M | 68.80 [41] |
| AutoBSS(ours) | 130M | 2.7M | 69.65 |

## 4.5 Generalization to Detection and Instance Segmentation

To investigate whether our method generalizes beyond the classification task, we also conduct experiments to search the BSS for the backbone of RetinaNet-R50 [42] and Mask R-CNN-R50 [43] on detection and instance segmentation task. We report results on COCO dataset [44]. As pointed out in [45] that ImageNet pre-training speeds up convergence but does not improve final target task accuracy, we train the detection and segmentation model **from scratch**, using SyncBN [46] with a 3x scheduler. The detailed settings are shown in Appendix B.4. The results are shown in Table 4. We can see that both $AP^{bbox}$ and $AP^{mask}$ are improved for Mask R-CNN with our searched BSS. $AP^{bbox}$ is improved by $0.91\%$ and $AP^{mask}$ is improved by $0.63\%$. Moreover, $AP^{bbox}$ for RetinaNet is improved by $0.66\%$ as well. This indicates that our method can generalize well beyond classification task.

Table 4: Comparison between the original BSS and the one searched by our method.

| Backbone | FLOPs | Params | $AP^{bbox}$ (%) | $AP^{mask}$ (%) |
|---|---|---|---|---|
| Mask R-CNN-R50 | 117B | 44M | 39.24 | 35.74 |
| Mask R-CNN-R50$^{AutoBSS}$ | 116B | 49M | 40.15 | 36.37 |
| RetinaNet-R50 | 146B | 38M | 37.02 | - |
| RetinaNet-R50$^{AutoBSS}$ | 146B | 41M | 37.68 | - |

## 4.6 Generalization for Searched BSSC to Similar Task

To investigate whether the searched BSSC can generalize to a similar task, we experiment on generalizing the BSSC searched for Mask R-CNN-R50 on instance segmentation task to semantic segmentation task. We report results on PSACAL VOC 2012 dataset [47] for PSPNet [48] and PSANet [49]. Our models are pre-trained on ImageNet and finetuned on train_aug (10582 images) set. The experiment settings are identical with [50] and results are shown in Table 5. We can see that both PSPNet50 and PSANet50 are improved equipped with the searched BSSC. It shows that the searched BSSC can generalize to a similar task.

Table 5: The single scale testing results on PSACAL VOC 2012.

| Method | mIoU (%) | mAcc (%) | aAcc (%) |
|---|---|---|---|
| PSPNet50 | 77.05 | 85.13 | 94.89 |
| PSPNet50$^{AutoBSS}$ | 78.22 | 86.50 | 95.18 |
| PSANet50 | 77.25 | 85.69 | 94.91 |
| PSANet50$^{AutoBSS}$ | 78.04 | 86.79 | 95.03 |

## 4.7 Qualitative Analysis for the Searched BSS

We further analyze the searched BSS configuration and give more insights. We compare the searched BSS with the original one in Figure 4. We can observe that the computation allocated uniformly for different stages in the original BSS configuration. This rule is widely adopted by many modern neural networks [4, 11, 8]. However, the BSS searched by AutoBSS presents a different pattern. We can observe some major differences from the original one:

1) The computation cost is not uniformly distributed, AutoBSS assign more FLOPs in latter stages. We think maybe the low-level feature extraction in shallow layers may not need too much computation, while the latter semantic feature may be more important.

2) AutoBSS increases the depth of early stages by stacking a large number of narrow layers, we think it may indicate that a large receptive field is necessary for early stages.

3) AutoBSS uses only one extremely wide block in the last stage, which may indicate that semantic features need more channels to extract delicately.

By the comparison of original BSS and the automatically searched one, we can observe that the human-designed principle for stacking blocks is not optimal. The uniform allocation rule can not make the most potential of computation.

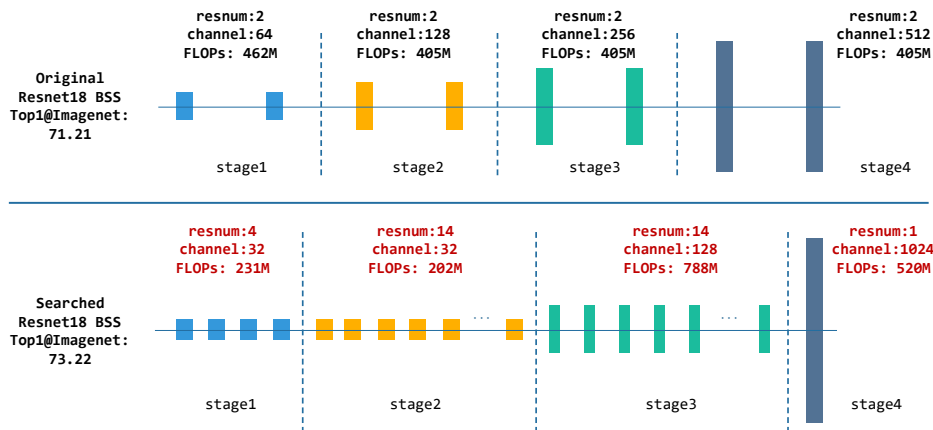

Figure 4: The difference between the original BSS and the searched one on ResNet18.

## 5   Conclusion

In this paper, we focus on the search of Block Stacking Style (BSS) of a network, which has drawn little attention from researchers. We propose a Bayesian optimization based search method named AutoBSS, which can efficiently find a better BSS configuration for a given network within tens of trials. We demonstrate the effectiveness and generalizability of AutoBSS with various network backbones on different tasks, including classification, model compression, detection and segmentation. The results show that AutoBSS improves the performance of well-known networks by a large margin. We analyze the searched BSS and give insights that BSS affects the computation allocation of a neural network, and different networks have different optimal BSS. This work highlights the impact of BSS on network design and NAS. In the future, we plan to further analyze the underlying impact of BSS on network performance. Another important future research topic is searching the BSS and block topology of a neural network jointly, which will further promote the performance of neural networks.

## Broader Impact

The main goal of this work is to investigate the impact of Block Stacking Style (BSS) and design an efficient algorithm to search it automatically. As shown in our experiments, the BSS configuration of current popular networks is not the optimal solution. Our methods can give a better understanding of the neural network design and exploit their capabilities. For the community, one potential positive impact of our work would be that we should not only focus on new convolutional operator or topological structure but also BSS of the Network. In addition, our work indicates that AutoML algorithm with unbiased evaluation has a strong potential for future research. For the negative aspects, our experiments on model compression may suggest that pruning based methods have less potential than tuning the BSS of a network.

## Acknowledgments and Disclosure of Funding

The authors thank the anonymous referees for their constructive feedback on the initial version of this paper. The authors also declare an absence of any competing interests.

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
