[Supplementary Material]

# A  Additional Details for AutoBSS

## A.1  Bayesian Optimization Based Search

In this procedure, we build a model for the accuracy of unevaluated BSSC based on evaluated one. Gaussian Process (GP, [1]) is a good method to achieve this in Bayesian optimization literature [2]. A GP is a random process defined on some domain $\mathcal{X}$, and is characterized by a mean function $\mu : \mathcal{X} \to R$ and a covariance kernel $\kappa : \mathcal{X}^2 \to R$. In our method, $\mathcal{X} \in R^m$, where m is the dimension of BSSC.

Given $\mathcal{O} = \{(x^{(0)}, y^{(0)}), (x^{(1)}, y^{(1)}), ...(x^{(n-1)}, y^{(n-1)})\}$, where $x^{(i)}$ is a refined BSSC for an evaluated BSS, $y^{(i)}$ is the corresponding accuracy. We firstly standardize $y^{(i)}$ to $\hat{y}^{(i)}$ whose prior distribution has mean 0 and variance $1 + \eta^2$ so that $\hat{y}^{(i)}$ can be modeled as $\hat{y}^{(i)} = f(x^{(i)}) + \epsilon_i$, here $f$ is a GP with $\mu(x) = 0$ and $\kappa(x, x') = exp(-\frac{1}{2\sigma^2}\|x - x'\|_{L_2}^2)$, $\epsilon_i \sim N(0, \eta^2)$ is the white noise term, $\sigma$ and $\eta$ are hyperparameters. Considering that the variance of $\epsilon_i$ should be much smaller than the variance of $\hat{y}^{(i)}$, we simply set $\eta = 0.1$ in our method. $\sigma$ determines the sharpness of the fall-off for kernel function $\kappa(x, x')$ and is determined dynamically based on the set $\mathcal{O}$, the details are illustrated below.

Firstly, we calculate the mean accuracy discrepancy for the BSSC evaluated in the first iteration. Because they are dispersed clustering centers, the mean discrepancy is relatively large. Then, we utilize this value to filter $\mathcal{O}$ and get the pairs of BSSC with a larger accuracy discrepancy than it. Afterward, we calculate the BSSC distance for the pairs and sort them to get $\{Dis_0, Dis_1, ..., Dis_{l-1}\}$, where $Dis_0 < Dis_1 < ... < Dis_{l-1}$. Finally, $\sigma$ is set as $\frac{Dis_{\frac{l}{20}}}{2}$.

It can be derived that the posterior process $f|\mathcal{O}$ is also a GP, we denote its mean function and kernel function as $\mu_n$ and $\kappa_n$ respectively. Denote $Y \in R^n$ with $Y_i = \hat{y}^{(i)}$, $k, k' \in R^n$ with $k_i = \kappa(x, x^{(i)}), k'_i = \kappa(x', x^{(i)})$, and $K \in R^{n \times n}$ with $K_{i,j} = \kappa(x^{(i)}, x^{(j)})$. Then, $\mu_n, \kappa_n$ can be computed via,

$$\mu_n(x) = k^T(K + \eta^2 I)^{-1}Y, \qquad \kappa_n(x, x') = \kappa(x, x') - k^T(K + \eta^2 I)^{-1}k'. \tag{1}$$

The value of posterior process $f|\mathcal{O}$ on each unevaluated BSS is a Gaussian distribution, whose mean and variance can be computed via equation 1. This distribution can measure the potential for each unevaluated BSSC. To determine the BSSC for evaluation, an acquisition function $\varphi : \mathcal{X} \to R$ is introduced, the BSSC with the maximum $\varphi$ value will be selected. There are kinds of acquisitions [3], we use expected improvement (EI) in this work,

$$\varphi_{EI}(x) = \mathbb{E}(max\{0, f(x) - \tau|\mathcal{O}\}), \tau = \max_{i < n}\left(\hat{y}^{(i)}\right). \tag{2}$$

EI measures the expected improvement over the current maximum value according to the posterior GP.

To reduce the time consumption and take advantage of parallelization, we train several different networks at a time. When selecting the first BSSC, equation 2 can be used directly. However, when selecting the following ones, there arises the problem that the accuracies for some BSSC are still unknown. Therefore, we use the expected value of EI function (EEI, [4]) instead. Supposing $x^{(n)}, x^{(n+1)}, ...$ are BSSC for selected BSS with unknown accuracies $\tilde{y}^{(n)}, \tilde{y}^{n+1}, ...$, thus

$$\varphi_{EEI}(x) = \mathbb{E}(\mathbb{E}(max\{0, f(x) - \tau|\mathcal{O}, \left(x^{(n)}, \tilde{y}^{(n)}\right), \left(x^{(n+1)}, \tilde{y}^{(n+1)}\right), ...\})), \tag{3}$$

here $\tilde{y}^{(n+j)}$ is a variable of Gaussian distribution with mean and variance depend on $\tilde{y}^{(n)}, ..., \tilde{y}^{n+j-1}$. The value of equation 3 is calculated via Monte Carlo simulations [4] in our method.

## B  Additional Details for Experiments

### B.1  Settings for the training networks on ImageNet

To study the performance of our method, a large number of networks are trained, including networks with original BSS and the ones with sampled BSS. The specifics are shown in Table 1

Table 1: Settings for the training networks on ImageNet.

| Shared | Optimizer: | SGD | Batch size: | 1024 |
| | Lr strategy: | Consin[6] | Label smooth: | 0.1 |
| | Weight decay | Initial Lr | Num of epochs | Augmentation |
| ResNet18 | 0.00004 | 0.4 | 120 | - |
| ResNet50 | 0.0001 | 0.7 | 120 | - |
| MobileNetV2 | 0.00002 | 0.7 | 120 | - |
| EfficientNet-B0 | 0.00002 | 0.7 | 350 | Fixed AutoAug[8] |
| EfficientNet-B1 | 0.00002 | 0.7 | 350 | Fixed AutoAug[8] |

### B.2  Additional Details for the Definition of BSSC

#### B.2.1  Definition of BSSC for ResNet18/50 and MobileNetV2

The definition of BSSC for EfficientNet-B0/B1 has been introduced in paper. Here we demonstrate the ones for ResNet18/50 and MobileNetV2.

Figure 1: The backbone of ResNets. $L_2, L_3, L_4, L_5$ are number of blocks for each stage, $C_2, C_3, C_4, C_5$ are the corresponding output channels.

ResNet18/50 consists of 6 stages as illustrated in Figure 1. Stages 1∼5 down-sample the spatial resolution of input tensor, stage 6 produces the final prediction by a global average pooling and a fully connected layer. The BSSC for ResNet18 is defined as the tuple $\{C_2, ..., C_5, L_2, ..., L_5\}$, where $C_i$ denotes the output channels, $L_i$ denotes the number of residual blocks. As for the BSSC of ResNet50, we utilize $B_i$ to describe the bottleneck channels for stage $i$, thus it becomes $\{C_2, ..., C_5, L_2, ..., L_5, B_2, ..., B_5\}$.

MobileNetV2 consists of 9 stages, its main building block is mobile inverted bottleneck MBConv [9]. The specifics is shown in Table 2. In our experiment, the BSSC for MobileNetV2 is defined as the tuple $\{C_3, ..., C_8, L_3, ..., L_8, T_3, ..., T_8\}$. $C_i$, $L_i$ and $T_i$ denote the output channels, number of blocks and expansion factor[9] for stage $i$ respectively.

#### B.2.2  Constraints on the BSSC

For the BSSC of ResNet18, we constrain $\{C_2, ..., C_5\}$ to be the power of 2, with the minimum as $\{2^5, 2^5, 2^6, 2^7\}$ and maxmum as $\{2^8, 2^{10}, 2^{11}, 2^{11}\}$. Moreover, the range of $L_i, i = 2, ..., 5$ is constrained to be $[1, 16]$. For the BSSC of ResNet50, we constrain $\{C_2, ..., C_5\}$ to be

Table 2: MobileNetV2 network. Each row describes a stage $i$ with $L_i$ layers, with input resolution $H_i \times W_i$, expansion factor[9] $T_i$ and output channels $C_i$.

| Stage $i$ | Operator $F_i$ | Resolution $H_i \times W_i$ | Expansion factor $T_i$ | Channels $C_i$ | Layers $L_i$ |
|---|---|---|---|---|---|
| 1 | Conv3x3 | $224 \times 224$ | - | 32 | 1 |
| 2 | MBConv,k3x3 | $112 \times 112$ | 1 | 16 | 1 |
| 3 | MBConv,k3x3 | $112 \times 112$ | 6 | 24 | 2 |
| 4 | MBConv,k3x3 | $56 \times 56$ | 6 | 32 | 3 |
| 5 | MBConv,k3x3 | $28 \times 28$ | 6 | 64 | 4 |
| 6 | MBConv,k3x3 | $14 \times 14$ | 6 | 96 | 3 |
| 7 | MBConv,k3x3 | $14 \times 14$ | 6 | 160 | 3 |
| 8 | MBConv,k3x3 | $7 \times 7$ | 6 | 320 | 1 |
| 9 | Conv1x1&Pooling&FC | $7 \times 7$ | - | 1280 | 1 |

the multiple of $\{64, 128, 256, 512\}$, with the minimum as $\{64, 128, 256, 512\}$ and maxmum as $\{320, 768, 1536, 3072\}$. $L_i, i = 2, ..., 5$ are constrained to be no larger than 11. The value of $B_i$ is chosen from $\{\frac{C_i}{8}, \frac{C_i}{4}, \frac{3C_i}{8}, \frac{C_i}{2}\}$ for stage $i$.

For the BSSC of MobileNetV2, we constrain $\{C_3, ..., C_8\}$ to be the multiple of $\{4, 8, 16, 24, 40, 80\}$, with the minimum as $\{12, 16, 32, 48, 80, 160\}$ and maxmum as $\{32, 64, 128, 192, 320, 640\}$. $\{L_3, ..., L_8\}$ are constrained to be no larger than $\{5, 6, 7, 6, 6, 4\}$. The value of expansion factor $T_i$ is chosen from $\{3, 6\}$.

For the BSSC of EfficientNet-B0/B1, we constrain $\{C_3, ..., C_8\}$ to be the multiple of $\{4, 8, 16, 28, 48, 80\}$, with the minimum as $\{16, 24, 48, 56, 144, 160\}$ and maxmum as $\{32, 64, 160, 280, 480, 640\}$. $\{L_3, ..., L_8\}$ are constrained to be no larger than $\{6, 6, 7, 7, 8, 5\}$. The value of expansion factor $T_i$ is chosen from $\{3, 6\}$.

## B.3 Additional Details for the Searched BSSC

### B.3.1 The Searched BSSC on ImageNet

Table 3: Comparison between the original BSSC and the one searched by random search or our method.

| Method | BSS Code | FLOPs | Params |
|---|---|---|---|
| ResNet18[10] | $\{64,128,256,512,2,2,2,2\}$ | 1.81B | 11.69M |
| ResNet18$^{Rand}$ | $\{32,64,128,512,1,2,8,5\}$ | 1.74B | 24.87M |
| ResNet18$^{AutoBSS}$ | $\{32,32,128,1024,4,14,14,1\}$ | 1.81B | 16.15M |
| ResNet50[10] | $\{256,512,1024,2048,3,4,6,3,64,128,256,512\}$ | 4.09B | 25.55M |
| ResNet50$^{Rand}$ | $\{128,640,1024,3072,8,7,4,3,48,80,256,384\}$ | 3.69B | 23.00M |
| ResNet50$^{AutoBSS}$ | $\{128,256,768,2048,9,6,9,3,48,128,192,512\}$ | 4.03B | 23.73M |
| MobileNetV2[11] | $\{24,32,64,96,160,320,2,3,4,3,3,1,6,6,6,6,6,6\}$ | 300M | 3.50M |
| MobileNetV2$^{Rand}$ | $\{20,48,80,144,200,480,2,4,3,2,5,1,3,3,3,6,3,3\}$ | 298M | 4.00M |
| MobileNetV2$^{AutoBSS}$ | $\{24,40,64,96,120,240, 4,3,2,3,6,2,3,6,6,3,6,6\}$ | 296M | 3.92M |
| EfficientNet-B0[11] | $\{24,40,80,112,192,320, 2,2,3,3,4,1,6,6,6,6,6,6\}$ | 385M | 5.29M |
| EfficientNet-B0$^{Rand}$ | $\{24,40,96,112,192,640, 1,2,1,2,6,1,6,6,6,6,3,6,6\}$ | 356M | 6.67M |
| EfficientNet-B0$^{AutoBSS}$ | $\{28,48,80,140,144,240, 3,1,4,2,6,3,3,3,6,3,6,6\}$ | 381M | 6.39M |
| EfficientNet-B1[11] | $\{24,40,80,112,192,320,3,3,4,4,5,2,6,6,6,6,6,6\}$ | 685M | 7.79M |
| EfficientNet-B1$^{Rand}$ | $\{16,24,128,140,240,400,4,6,2,4,5,2,3,3,3,3,6,6\}$ | 673M | 10.19M |
| EfficientNet-B1$^{AutoBSS}$ | $\{24,64,96,112,192,240,3,1,3,4,7,5,6,3,3,3,6,6\}$ | 684M | 10.17M |

### B.3.2 The Searched BSSC on Model Compression

Table 4: Comparison between the BSSC obtained by uniformly rescaling or AutoBSS for MobileNetV2.

| Method | BSS Code | FLOPs | Params |
|---|---|---|---|
| Uniformly Rescale | {16,20,40,56,96,192,2,3,4,3,3,1,6,6,6,6,6,6} | 130M | 2.2M |
| AutoBSS(ours) | {16,20,40,56,96,288,1,4,6,1,6,1,3,6,6,3,6,6} | 130M | 2.7M |

### B.3.3 The Searched BSSC on Detection and Instance Segmentation

Table 5: Comparison between the original BSS and the one searched by AutoBSS.

| Backbone | BSS Code | FLOPs | Params |
|---|---|---|---|
| Mask R-CNN-R50 | {256,512,1024,2048,3,4,6,3,64,128,256,512} | 117B | 44M |
| Mask R-CNN-R50$^{AutoBSS}$ | {192,512,512,1024,9,3,6,10,24,192,256,384} | 116B | 49M |
| RetinaNet-R50 | {256,512,1024,2048,3,4,6,3,64,128,256,512} | 146B | 38M |
| RetinaNet-R50$^{AutoBSS}$ | {192,384,768,1536,3,7,9,8,72,192,192,384} | 146B | 41M |

## B.4 Additional Details for Settings on Detection and Instance Segmentation

We train the models on COCO [12] train2017 split, and evaluate on the 5k COCO val2017 split. We evaluate bounding box (bbox) Average Precision (AP) for object detection and mask AP for instance segmentation.

**Experiment settings** Most settings are identical with the ones on ImageNet classification task, we only introduce the task specific settings here. We adopt the end-to-end fashion [13] of training Region Proposal Networks (RPN) jointly with Mask R-CNN. All models are trained from scratch with 8 GPUs, with a mini-batch size of 2 images per GPU. We train a total of 270K iterations. SyncBN [14] is used to replace all 'frozen BN' layers. The initial learning rate is 0.02 with 2500 iterations to warm-up [5], and it will be reduced by $10\times$ in the 210K and 250K iterations. The weight decay is 0.0001 and momentum is 0.9. Moreover, no data augmentation is utilized for testing, and only horizontal flipping augmentation is utilized for training. The image scale is 800 pixels for the shorter side for both testing and training. The BSSC is defined as the one for ResNet50 on ImageNet classification task and we also constrain the FLOPs for the searched BSS no larger than the original one. FLOPs is calculated with an input size $800 \times 800$. Only backbone, FPN and RPN are taken into account for the FLOPs of Mask R-CNN-R50. In the searching procedure, we only target to search BSS for higher AP.