[Reviews · NeurIPS 2020]

Review 1

Summary and Contributions: - Authors study and use a method to figure out the optimal way to stack network "blocks" (BSS) for a variety of different networks. They change the number of blocks and the channels for each block in the network. - The authors use bayesian optimization for their search method for finding the optimal block stacking order once they apply a clustering method to make the search more efficient - They try this on a variety of different architectures on ImageNet and get good improvements. They also try on a variety of other tasks like object detection, instance segmentation and model compression.

Strengths: - The authors did a nice job of running random baselines compared to their method, to show that their search method is necessary - Over all the paper addresses an important and often overlooked topic when designing neural network architectures. It is quite practical and is additive to architectures people have already discovered. - The method works on lots of tasks and models, some of which are strong baselines - The authors do a nice job of running ablations for their algorithm (Table 2) - Due to the authors method, the computation complexity is not high - The algorithm the authors use is novel and makes sense to use given their analysis in the paper

Weaknesses: - In Table 1, how do the latencies change on devices like GPU as FLOPS can be misleading in how it translates to runtime performance. - Overall the improvements are not that large on most architectures and on some of the setups the baselines are not that strong (Table 4 both RetinaNet and Mask RCNN). - In Table 1 it would be nice to compare to prior BSS search methods like POP [23] - It would be nice to have a figure detailing an example BSSC to make it more intuitive to the reader in the main text - More details can be given for some of the plots as they are hard to understand from just looking at them with the given caption (EX: in 3(b) explain what is going on more thoroughly. In 3(a) what is the x-axis?)

Correctness: Yes the claims, methods and empirical methodology is correct.

Clarity: - The paper is written moderately well, but there are a fair bit of typos scattered throughout the text and certain parts are unclear - I found 3.1 to be unclear and took a while to understand - Examples: - Line 108: "Benefit from this hypothesis" -> "To benefit from this hypothesis" - Line 128: "It can be proved in random sampling", what can be proved "in" random sampling?

Relation to Prior Work: Yes the paper does a good job of framing their work vs prior work

Reproducibility: Yes

Additional Feedback:


Review 2

Summary and Contributions: This paper proposes a novel neural architecture search method based on the optimization of the configuration of predefined blocks. The main contribution of the paper is the definition of a search space, a coding, which results in strong performance with very few evaluations. Other contributions include a first optimization step called refining, and a filtering of candidate points based on clustering, then fed into a classical Bayesian optimization method. Some improvements are demonstrated on state of the art models, results also suggest that optimal block stacking might be narrower in the first stages and larger in the later stages. Edit post rebuttal: I think the authors did a reasonable job in addressing my questions. My biggest complaint was the poor justification of the BSSC, but with the additionnal explanation and experiment I feel confident it is a beneficial addition. I amended my score.

Strengths: The strengths of this paper are methodological and experimental. The proposed method seems to improve upon the state of the art. Given the relative maturity of block-based classification models, gains are often in the few percentage points or lower, but I would argue this is still significant. Some authors have studied Block stacking approaches, but computational budgets seem to be significantly higher (no direct comparison provided however).

Weaknesses: Even though I find the proposed approach interesting and I think it can have impact in the field, there are significant issues in the justification and perhaps the soundness of the method. Some steps of the method are not justified in a convincing manner, perhaps some further experiments might help with this. Some key elements are missing to get a clear understanding of the method (this ties in to clarity). For instance, the authors state (page 4, lines 133-137): "Specifically, we first randomly select one dimension i of BSSC, then BSSC_i will be increased by a predefined step size if this doesn’t result in larger FLOPs or latency than the threshold. This process will be looped until no dimension can be increased." This seems to assume that increasing the parameter / number of blocks (at this point it's still not clear what the parameters refer to) will necessarily improve the performance of the resulting network. Is that a verified assumption? More importantly, I could not understand exactly what is the purpose of BSSC refining -- it should be better explained and justified. The proposed one-linear neural network takes as input a BSSC and outputs a new BSSC that has a lower euclidean distance if accuracies are similar. How does this modification of the BSSC make sense in the parameter space? Why does lowering the Euclidean distance of points result in better candidate suggestions? How is the one-layer network initialized when starting the optimization? Is it pre-learned on other datasets? Why did you do an ablation study for the clusters but not the refining? Some detail is also lacking in order to be able to reproduce this work. For the Bayesian optimization part, how is the acquisition function optimized? What is the set of candidates used for the optimization, the set of k-mean cluster centers (not explicited anywhere in the paper)? Some points mentioned above are also important with regards to reproducibility.

Correctness: The method may or may not be correct, depending on how the points above are addressed.

Clarity: The first 4-5 pages of the paper are hard to digest and could benefit from proofreading. There are numerous grammar mistakes. Find some comments on clarity below, but the paper probably needs to be rewritten. The name block-wise structure and Block stacking style are confusing (at least to this reviewer). They sound like they refer to the same thing. I suggest replacing "block-wise structure" with something else. Block-wise structure does not imply that the structure of the block is modified, rather the structure with regards to the blocks, i.e. the layout of blocks. It's not a major point I think it would help with adoption of your method. line 118: what is R? are all x_i's are in the same set of values? line 119: the sentence "|·| denotes the number of elements for a set" is unneeded in this paragraph. Section 3.2: What is the definition of accuracy discrepancy for Figure 2? Discrepancy can be defined in many ways, it seems important to include the equation used or at the very least a reference. Table 2 header: clustring -> clustering Figure 3b) should also reproduce the original relationship between accuracy and unrefined BSSC for easier comparison.

Relation to Prior Work: To my knowledge the authors correctly differentiate their contributions from that of previous works.

Reproducibility: Yes

Additional Feedback: I think the authors should add an ablation study on the refining process. It would also help to gain a better understanding to study the way in which BSSCs are influenced by the refining step, as well as more information on how it is trained.


Review 3

Summary and Contributions: The authors propose a novel NAS method that searches the Block Stacking Style (BSS). Different from most NAS methods that search operations and topology of a block, this paper focuses on searching the computation allocation of a network, namely searches the width and depth of a network for a given block. I think this paper gives a new perspective to NAS community that, how to effectively stack the building block of networks has a big impact on the importance. Overall, this paper is well written and the idea is novel, experiment results on classification, detection, segmentation and model compression demonstrate the effectiveness and generalization of proposed method.

Strengths: 1. The main idea of studying the computation allocation by searching the Block Stacking Style is novel, and may give a new perspective to NAS community. The proposed method is also complementary to most block based NAS methods. 2. The proposed Bayesian based search algorithm is sample efficient, and does not involve biased tricks usually used in NAS methods, such as early stop or weight sharing.

Weaknesses: 1. Some existing NAS methods can also search the width and depth of network, such as single path one shot, DARTs, more discussions of these methods should be added. 2. The results are under FLOPs constraints, I wonder if the proposed AutoBSS can be used under latency constraints.

Correctness: correct

Clarity: Clearly written

Relation to Prior Work: Since this paper focuses on searching widths and depth, it is better to specifically discuss the connection between the proposed method and existing NAS methods that can search width and depth of networks, such as single path one shot, darts, EfficientNet.

Reproducibility: Yes

Additional Feedback: After reading the rebuttal and comments from other reviewers, I would like to keep my original rating. Different from most NAS methods that search operations and topology of a block, this paper focuses on searching the computation allocation of a network, which is meaningful to improve the hardware utilization for AI chip.


Review 4

Summary and Contributions: This paper proposes a Bayesian optimization based search method. It can find optimal block stacking style tens of trials. The experiments demonstrate the effectiveness of the proposed method.

Strengths: The idea of designing Block Stacking Style is interesting. The experiments on image classification, object detection, instance segmentation and model compression demonstrates the generalizability of the proposed method. The paper is well written and easy to follow in general.

Weaknesses: 1. From the detectron2 model zoo (https://github.com/facebookresearch/detectron2/blob/master/MODEL_ZOO.md). I find that RetinaNet-R50 with 3x learning scheduler achieves 38.7. In the Tab.4, the result is 37.02. I would like to know why there is a big gap between the model zoo results and your results. Mask R-CNN is the same. 2. In Table 1, I observe that you conduct experiment on EfficientNet-B0 and EfficientNet-B1. Why donot you conduct experiments on EfficientNet-B7 or some stronger network?

Correctness: The method and empirical methodology are correct.

Clarity: This paper is well written.

Relation to Prior Work: It is clear that difference between this paper and previous works states clearly.

Reproducibility: Yes

Additional Feedback: The k-means are adopted in your method. Do you try other clustering methods? How long is the training time? And what is the number of GPUS.? Do you have any plan to make your source code and models pubulic? I would be happy to see the results on pixel-level applications such as, semantic segmentation.

[Author Response · NeurIPS 2020]

We appreciate the reviewers for the constructive comments on this paper. We will carefully address the grammar
mistakes, undetailed plots, unclear statements in the revised manuscript. Major concerns are responded as follows:
**One common concern is our baseline for RetinaNet/Mask-RCNN is not strong.** It is because our models are
trained from scratch (Section 4.5), while most results of other papers or model zoo are fine-tuned from a pre-trained
model. The result of Mask-RCNN with 3x scheduler and training from scratch in "Kaiming He, et al. Rethinking
ImageNet Pre-Training." is comparable with our baseline (39.5% vs 39.24%).

**R1Q1: How do the latencies change on GPU?** We show the latencies change in Table 1, which indicates models of
similar FLOPs vary greatly in latency, e.g. ResNet18$^{AutoBSS}$ is **18% slower** while Effi-B1$^{AutoBSS}$ is **23% faster**.

Table 1: Average GPU latency per image (Tesla V100-PICe, Pytorch 1.2.0, CUDA 9.0.167, CUDNN 7102).

| ResNet18 | Latency | ResNet50 | Latency | MBV2 | Latency | Effi-B0 | Latency | Effi-B1 | Latency |
|---|---|---|---|---|---|---|---|---|---|
| Original | 0.242ms | Original | 0.973ms | Original | 0.257ms | Original | 0.438ms | Original | 0.703ms |
| Rand | 0.200ms | Rand | 1.22ms | Rand | 0.206ms | Rand | 0.346ms | Rand | 0.544ms |
| AutoBSS | 0.287ms | AutoBSS | 1.11ms | AutoBSS | 0.252ms | AutoBSS | 0.407ms | AutoBSS | 0.539ms |

**R1Q2: The improvements are not large.** As pointed out by Reviewer#2, the baseline networks are relatively mature
thus the improvements are still significant. We think this is especially the case on common datasets like ImageNet, the
improvements will be larger on other datasets.

**R1Q3: Compare to prior BSS search methods like POP[22] in Table 1.** The network "DF2" in POP has similar
FLOPs (1.77G vs 1.8G) and Top1-Acc (73.9% vs 73.91%) compared with ResNet18$^{AutoBSS}$. But their searching is
more sample inefficient (200 vs 64 samplings) and they need some handcrafted modifications like changing the kernel
size for the first convolution from $7 \times 7$ to $3 \times 3$. Methodological comparison can be found in line $90 \sim 93$.

**R2Q1: Why increasing one dimension of BSSC necessarily make the resulting network perform better than the**
**original network?** Because the resulting network strictly includes the original one, thus has a greater capacity. Take
MobileNetV2[7] as an example. Increasing one dimension of BSSC means increasing the number of channels or
blocks. For channels, a larger width multiplier always results in better performance (table 4 of [7]). For blocks, residual
connection guarantees the newly-introduced inverted residual block no worse than identity mapping.

**R2Q2: The purpose of BSSC refining.** 1) Bayesian Optimization approach based on a Lipschitz-continuous
assumption (page 4 of [33]). The refining makes the model for accuracy, i.e. $f(x)$ satisfy this assumption better. That is,
there exists constant $C$, such that for any two BSSC $x_1, x_2$: $\|f(x_1) - f(x_2)\| \leq C\|x_1 - x_2\|$. The red dashed line in
Figure 2 and Figure 3(b) (mean value plus standard deviation) can be regarded as the upper bound of $\|f(x_1) - f(x_2)\|$.
After refined with a linear layer, it becomes much more close to the form of $C\|x_1 - x_2\|$. 2) It helps to aggregate
the BSSC with similar accuracies into one cluster, thus they will not be sampled at the same time, otherwise, some
evaluations for BSSC will be meaningless.

**R2Q3: Some unclear details. A) How is the one-layer network initialized?** The initial weight is set as an identity
matrix (line 158∼159). **B) Is the one-layer network pre-learned on other datasets?** No, the linear layer for Figure
3(b) is trained with 16 BSSC, thus the tens of BSSC evaluated during each searching procedure is enough to train this
model. **C) How is the acquisition function optimized and what is the set of candidates for this optimization?**
We simply calculate the value of acquisition function at each clustering center and select the largest one (line 164∼165).
**D) What is the definition of accuracy discrepancy?** The absolute difference of two accuracies.

**R2Q4: Add an ablation study on the refining process.** Actually, we have conducted experiments on whether to
refine BSSC with a linear model. Without this step, the accuracy for the searched ResNet18$^{AutoBSS}$ is $\sim 0.4\%$ lower.
We thought the comparison between Figure 2 and Figure 3(b) could indicate the importance of refining thus didn't
report this ablation result, we will add it for the revision.

**R3Q1: Add more discussions of existing NAS methods like DARTs.** These methods rely on a biased evaluation
protocol like early stop or parameter sharing, while we can train each sampled network fully and get an unbiased
evaluation. More details about algorithm efficiency and effectiveness will be added in the revision.

**R3Q2: If the proposed AutoBSS can be used under latency constraints?** Our method supports other constraints
like CPU/GPU latency. It can be achieved via using latency instead of FLOPs to construct the candidate set.

**R4Q1: Why don't you conduct experiments on EfficientNet-B7 or some stronger network?** Actually,
EfficientNet-B0/B1 is already a very strong network under the constraint of FLOPs, thus can verify our method.
We didn't use EfficientNet-B7 because it is too resource-consuming.

**R4Q2: Do you try other clustering methods?** Haven't yet. Thanks for your reminder, we will try it for future work.

**R4Q3: How long is the training time? And what is the number of GPUs?** We take ResNet18 as an example. It
needs $\sim 20$ hours to train a model with 8 GPUs. For each iteration, the 16 sampled models are trained in parallel.

**R4Q4: Do you have any plan to make your source code and models pubulic?** Yes, we have applied for sharing
the code and models for academic society, it is in the approval process.

**R4Q5: Results on semantic segmentation.** We conduct experiments on PSACAL VOC 2012 for PSPNet and
PSANet. The training settings are identical with *Performance.table2* of the github project (hszhao/semseg) created by
authors of PSPNet and PSANet. We pre-train our model on ImageNet as well. Because of the task similarity of semantic
segmentation and instance segmentation, we directly adopt the BSS searched for the backbone of Mask-RCNN-R50.
By adjusting the BSS of backbone, we improve the **mIoU/mAcc/aAcc** (single scale testing) from **77.05/85.13/94.89%**
to **78.22/86.50/95.18%** for PSPNet50 and from **77.25/85.69/94.91%** to **78.04/86.79/95.03%** for PSANet50.

[Meta-Review · NeurIPS 2020]

This paper initially got mixed recommendations, three positive and one negative. The reviewers agree that this paper addresses an important problem for neural network architecture design. The experiments are comprehensive and results are good. However, one reviewer has the concerns on the experimental justification and that gave a weak reject. This concern was addressed by the additional experiments in the authors' response. Finally all the reviewers agree for acceptance. AC concurs.